# Presence of Non-Tuberculous Mycobacteria Including *Mycobacterium avium* subsp. *paratuberculosis* Associated with Environmental Amoebae

**DOI:** 10.3390/ani13111781

**Published:** 2023-05-27

**Authors:** Vincent Rochard, Thierry Cochard, Stéphanie Crapart, Vincent Delafont, Jean-Louis Moyen, Yann Héchard, Franck Biet

**Affiliations:** 1Laboratoire Ecologie et Biologie des Interactions, UMR Centre National de la Recherche Scientifique 7267, Université de Poitiers, Equipe Microbiologie de l’Eau, F-86073 Poitiers, Francevincent.delafont@univ-poitiers.fr (V.D.); yann.hechard@univ-poitiers.fr (Y.H.); 2Institut National de Recherche pour l’Agriculture—INRAE, Université de Tours, ISP, F-37390 Nouzilly, France; thierry.cochard@inrae.fr; 3Laboratoire Départemental d’Analyse et de Recherche de la Dordogne, F-24660 Coulounieix-Chamiers, France

**Keywords:** amoebae, environment, Johne’s disease, *Mycobacterium avium* subsp. *paratuberculosis*, transmission

## Abstract

**Simple Summary:**

The prevalence of Johne’s disease (JD) due to *Mycobacterium avium* ssp. *paratuberculosis* (*Map*), responsible for considerable economic losses in the dairy industry worldwide, remains very high. In addition, recent studies have shown that *Map* may hamper the success of bovine tuberculosis (bTB) eradication programs due to frequent co-infection of hosts. Therefore, the risk of *Map* contamination by environmental sources must be studied given achieving improved control of this enzooty. Our previous work showed that environmental amoebae could function as reservoirs and vectors of these pathogenic mycobacteria. This study aims to describe amoeba-mycobacteria co-occurrence by sampling the water points of cattle herds monitored for *Map* and *Mycobacterium bovis* (*M. bovis*) infection. Our study showed that a wide diversity of amoebae and non-tuberculous mycobacteria (NTM) species, including *Map*, live closely together in water troughs of herds monitored for JD or bTB. The exact association between amoebae and mycobacteria calls for further investigation.

**Abstract:**

One of the obstacles to eradicating *paratuberculosis* or Johne’s Disease (JD) seems to be the persistence of *Mycobacterium avium* subsp. *paratuberculosis* (*Map*) in the environment due to its ability to survive alone or vectorized. It has been shown that *Map* is widely distributed in soils and water. Previously, we isolated amoebae associated with *Map* strains in the environment of bovines from an infected herd. This work aims to verify our working hypothesis, which suggests that amoebae may play a role in the transmission of JD. In this study, we sampled water in the vicinity of herds infected with *Map* or *Mycobacterium bovis* (*M. bovis*) and searched for amoebae and mycobacteria. Live amoebae were recovered from all samples. Among these amoebae, four isolates associated with the presence of mycobacteria were identified and characterized. *Map* and other mycobacterial species were detected by qPCR and, in some cases, by culture. This study suggests that amoebae and *Map* may be found in the same environment and might represent a risk of exposure of animals to pathogenic mycobacteria. These data open up new perspectives on the control measures to be put in place to prevent contamination by *Map*.

## 1. Introduction

Johne’s disease (JD), caused by *Mycobacterium avium* subsp. *paratuberculosis* (*Map*) is a chronic ruminant disease responsible for considerable economic losses in the dairy industry worldwide [1]. Despite implementing control programs in most developed countries and notwithstanding substantial financial efforts, the prevalence of Johne’s disease remains very high (around 50% in European herds and 80% in the United States) [2,3].

The reasons for this worrisome situation and its negative impact on animal health and the livestock economy are multiple, both in terms of the biology of the disease (especially due to the long-term sub-clinical nature of the infection) and of the specificities of this very slowly growing mycobacteria, of which the isolation always remains a challenge, as do control policies [4].

Another concern is that *Map* may hamper bovine TB (bTB) eradication programs due to frequent co-infection of *Map* and *M. bovis*. This may lead to misdiagnosis (preventing accurate identification of infected animals), and it has been suggested that herds with “chronic” *M. bovis* infection may be more likely to present with co-infection [5]. A key factor in the difficulty of controlling these infections may be the environmental survival of these mycobacteria.

Cattle infected with *Map* shed the bacteria in their faeces. For this reason, fecal shedding by cows is the main route of transmission to calves [6]. Simulation models have confirmed the hypothesis that intermittent, low, and transient shedding animals play a major role in maintaining a low prevalent infection rate in dairy herds. As a result, *Map* is widely distributed within and beyond the boundaries of the agricultural environment, in both soil and water [7]. However, these analyses are often based on the detection of DNA and do not provide information on the general potential for survival and multiplication of ex-host *maps*, which is a key element in the dynamics of disease transmission.

Microbial hosts in the shared agricultural environment may likely serve as vectors to *Map* and contribute to the spread and persistence of JD. Environmental protozoa such as free-living amoebae (FLA) have long been considered “incubators” for intracellular bacteria in the environment [8,9]. They are a training ground for pathogenic bacteria, including several *Mycobacterium* species [10,11,12]. The interactions between *Map* and amoebae remain little studied to date. A previous study involving amoebae has shown that *Acanthamoeba* spp. It enables *Map* replication and enhances virulence [13]. Another study showed that *Map* ingested by *Acanthamoeba* resisted digestion for at least 24 days [14]. White et al. described the co-occurrence of *Map* and FLA in soil (2010) [15]. An in vitro study demonstrated that *Map* persisted for up to four years in the presence of *Acanthamoeba* [16].

Interestingly, a report showed that *Map* was found in *amoebae* isolated from the soil after the application of cattle manure spiked with *Map* [17]. In our previous study, we reported that various lineages of *Map* strains were able to grow within *Acanthamoeba castellanii* and that they can survive for several days within their host [18]. In addition, we showed that *Map* might be found in amoebae isolated from a cattle environment, and we also showed that the strain genotype was identical to a previous infectious strain isolated from a bovine [18]. These results strongly suggest a risk of transmission mediated by infected amoebae.

In the present study, we sought to extend these previous investigations on amoeba-mycobacteria association by sampling through water points from six herds monitored for *Map* and *M. bovis* infection status.

## 2. Materials and Methods

### 2.1. Sampling

Sampling campaigns occurred from April 2018 to April 2019 in six farms in southwestern France (Figure 1 and Table 1). For this study, we selected farms from this area for two reasons. The first is that JD is endemic in this region, and the second is that although France is one of the officially tuberculosis-free (OTF) member states of the EU, it nevertheless experiences multi-host tuberculosis enzootic situations, mainly in the southwestern region [19]. The bTB and JD status of the herds enrolled in this study is detailed in Table 1. On average, three samples were taken per herd, including troughs and outdoor water points. Figure 2 summarizes the process of sample treatment.

### 2.2. Cultivation of Microorganisms

Mycobacterial strains were grown at 37 °C in Sauton medium or Middlebrook 7H9 broth as described previously [20].

*Escherichia coli* strain K12 was used to isolate amoeba as described previously [18].

### 2.3. Isolation of Environmental Amoebae

The samples, comprising two flacons of one liter, were taken from the drinking troughs for each herd. They were treated according to the method described by Samba-Louaka et al. [18]. Briefly, water samples were filtered, and the filters were placed on agar plates seeded with *E. coli* as a food source. The plates were incubated at 20 or 37 °C, and the presence of amoeba development was followed by microscopy. To obtain a clonal culture of amoebae, they were subcultured in the same conditions or liquid culture: PAS buffer [18] spiked with *E. coli* in a 25 or 75 cm^2^ flask.

### 2.4. DNA Extraction and Purification

The environmental samples were centrifuged at 13,000× *g* for 20 min at +4 °C. DNAs were extracted from 500 mg of pellet using the FastDNA Spin Kit for Soil, MP Biomedicals (67400 Illkirch-Graffenstaden, France) according to the manufacturer’s instructions. The obtained DNAs were stored until use at −20 °C.

### 2.5. Detection and Quantification of Mycobacteria by Quantitative PCR (qPCR)

Bacterial genomic DNA was detected by quantitative PCR (qPCR) in a final volume of 25 µL including 5 µL of extracted DNA, 12.5 µL of TaqManTM Fast Advanced Master Mix with UNG (Applied Biosystems, Thermo Fischer Scientific, Vilnius, Lithuania), 2.5 µL of primer mix (Table 2) at 3 µM each and probed at 2.5 µM (table below) and 5 µL of ultrapure water. Experiments were carried out using a BIO-RAD CFX96 thermocycler (BIO-RAD), consisting of polymerase activation at 50 °C for 2 min, followed by an initial denaturation step at 95 °C for 10 min, followed by 40 cycles of denaturation at 95 °C for 15 s, elongation at 60 °C for 1 min. The Ct is calculated with the CFX Manager software (BIO-RAD CFX Manager 3.1). The primers used are listed in Table 2.

### 2.6. Identification of Mycobacteria Isolated from Samples

To isolate mycobacteria, the samples were decontaminated according to two methods in parallel. The first decontamination method used 0.9% hexadecylpyridinium chloride (HPC) (weight/volume). The environmental samples were centrifuged at 13,000× *g* for 20 min at +4 °C, and approximately 2 g of the pellet was resuspended in 25 mL of 0.9% HPC and incubated for 18 h at room temperature. The second decontamination method used 3% sodium dodecyl sulfate (SDS). Approximately 2 g of the pellet was resuspended with 30 mL of a 1% sodium hydroxide solution, 3% SDS and 25 mg/L phenol red and incubated at 37 °C for 30 min with regular gentle agitation. The suspension was then neutralized with H3PO4 until the color indicator turned.

After centrifugation for 15 min at 5000× *g* of suspensions obtained from the two decontamination methods, the pellets were resuspended with 2 mL of Middlebrook 7H9 broth (Difco Laboratories, Detroit, MI, USA) with 0.2% glycerol and supplemented with 10% Albumin Dextrose Catalase (ADC, Becton Dikinson, Le Pont de Claix, France) to inoculate two Herrold’s tubes with mycobactin J and ANV (Becton Dickinson, Le Pont de Claix, France). The tubes were incubated horizontally for one week and then vertically until cultures appeared (in a maximum of 16 weeks).

Mycobacterial species were identified as previously described [21].

### 2.7. Amoebal Identification

The amoeba isolated from the water troughs was identified as described previously [22]. Briefly, DNA from the amoebae was isolated and amplified by PCR with universal primers targeting the 18S rRNA gene (Table 2). Then, the amplified fragments were sequenced and compared to the nucleotide database using the BLAST program.

**Table 2 animals-13-01781-t002:** Primers used for mycobacteria and amoeba species identification.

Name	Primers	Sequences	Target
qPCR atpE	Forward	CGGYGCCGGTATCGGYGA	*Mycobacterium* ssp.
	Reverse	CGAAGACGAACARSGCCAT	
	Probe	^FAM^-ACSGTGATGAAGAACGGBGTRAA-^BHQ1^	
qPCR IS900	Forward	CCGCTAATTGAGAGATGCGATTGG	
	Reverse	AATCAACTCCAGCAGCGCGGCCTCG	*Map*
	Probe	^FAM^-TCCACGCCCGCCCAGACAGG-^BHQ1^	
	Forward	GGTAGCAGACCTCACCTATGTGT	
qPCR IS6110	Reverse	AGGCGTCGGTGACAAAGG	MTB complex *
	Probe	^FAM^-CACGTAGGCGAACCC-^MGB^	
qPCR IS1081	Forward	CCGCCACCGTGATTTCGA	
	Reverse	GCCAGTCCGGGAAATAGCT	MTB complex *
	Probe	^FAM^-CCGCAACCATCGACGTC-^MGB^	
qPCR IS1561	Forward	GATCCAGGCCGAGAGAATCTG	
	Reverse	GGACAAAAGCTTCGCCAAAA	MTB complex *
	Probe	^FAM^-ACGGCGTTGATCCGATTCCGC-^TAMRA^	
hsp65	Forward	CTTGTCGAACCGCATACCCT	
	Reverse	ACCAACGATGGTGTGTCCAT	*Mycobacterium* ssp.
18S rRNA	Forward 566	CAGCAGCCGCGGTAATTCC	Amoeba
	Reverse 1200	CCCGTGTTGAGTCAAATTAAGC	

* Detection of the MTBC complex targeted two specific insertion sequences: IS6110 and IS1081. This reflects the extra sensitivity that these multicopy sequences allow. In addition, the PCR targeting IS1561 was used to discriminate members of MTB to *Mycobacterium microti* [23].

## 3. Results

### 3.1. High Diversity of Non-Tuberculous Mycobacteria from Herd Samples

We attempted to detect mycobacteria’s indirect and direct presence in each sample according to the method illustrated in Figure 2. Table 3 summarizes the results of detecting DNA mycobacteria by analyses of five different qPCRs, identifying either the species or the subspecies of mycobacteria, and the results of the isolation and genetic identification of mycobacteria by culture. Overall, we observed that positive qPCR results were not confirmed in positive cultures, and conversely, samples leading to positive cultures were not always positive by qPCR analysis. Regarding qPCR analysis, the samples from herds 1, 3 and 4 did not give convincing results with signals above the positivity thresholds of Ct 40. For herd 2, only one sample was positive for *Mycobacterium* spp. For herds 5 and 6, many samples were positive for mycobacteria, including the detection of *Map* and species of the MTBC.

Bacteriological analyses were not consistent with qPCR data. For the samples from herds 1 and 2, which were almost negative in qPCR, widely diverse NTM, including *Map*, was isolated. Conversely, despite positive qPCR signals for mycobacteria in herds 5 and 6 samples, few mycobacteria could ultimately be isolated in culture, mainly due to fungal contaminants that are difficult to eliminate in environmental samples.

In these samples taken, it is interesting that *Map* was frequently found by qPCR and even in culture. Other interesting results of this study highlight the diversity of NTM species presents in these samples from troughs used by cattle.

### 3.2. Herd Water Samples Rich in Environmental Amoebae

In a previous study, we succeeded in isolating and culturing amoebae from water samples from cattle drinkers on a farm infected with *Map* [18]. In addition, the total DNA extracted from the culture of the amoebae demonstrated the presence of *Map* DNA. These encouraging results [18], prompted us to extend more widely these investigations aimed at isolating amoebae capable of harboring pathogenic mycobacteria.

By sampling six cattle herds, we were able to collect and analyze about twenty collection points. In most samples, the amoeba culture was positive at 20 or 37 °C (Table 4).

To identify the amoeba isolates, strains were sub-cultured, and PCR targeting a portion of the 18S rRNA coding gene was performed. Sequencing of purified amplicon made it possible to identify the amoeba species. A total of 22 amoebal cultures allowed the identification of five different genera of amoeba, including, in order of frequency in the samples, *Naegleria* (59%), *Acanthamoeba* (18%), *Vermamoeba* (14%), *Filamoeba* (4.5%) and Tubulinea group (4.5%) (Figure 3). Figure 4 shows the morphological aspects of the main amoebic species isolated during this study. Positive *atpE* qPCR analyses performed on these water samples suggest an association of these amoebae isolated from trough water points with mycobacteria.

## 4. Discussion

Livestock and wildlife spread pathogenic mycobacteria such as *Map* and *M. bovis* to pasture, soil, feed, water, and microfauna [24,25]. Field and experimental reports have indicated the persistence of these pathogenic mycobacteria under different environmental conditions representing a potential risk of infection [26].

The present study addressed the question of the presence of viable mycobacteria in the cattle environment, including environmental actors represented by free-living amoebae, protists frequently found in water and soils, to understand better the transmission routes of the disease among livestock and wildlife. For this purpose, we collected water samples from the environment of animals from six herds in southwest France. We looked for the presence of mycobacteria associated with the isolation of amoebae. The results of our study describe the presence, in all samples, of amoeba of the different genera related to positive qPCR signals of mycobacteria. Furthermore, the same samples revealed the presence of mycobacteria by qPCR and cultures, leading to the identification of different NTM species, including *Map*.

Detecting mycobacteria in environmental samples of waters containing highly diverse microbial communities, including several described and potentially undescribed NTMs [27], is always a challenge. That is why we chose two approaches, one based on detection by qPCR and the other based on the culture of mycobacteria. qPCR detection is a method of choice for mycobacteria, as it is a faster alternative to culture-based methods, hampered by the slow growth rate of mycobacteria and the high number and diversity of environmental species [28]. Additionally, PCR can detect mycobacterial DNA in environmental samples after initial contamination [29]. However, our results show that some samples, mainly from herds 1 and 2, did not show mycobacteria detection PCR signals, while at other sampling sites, we could detect mycobacteria signatures. These results could reflect different dilution rates of target mycobacteria in the samples or PCR inhibition by co-extracted contaminants. According to some studies, these effects could be minimized using an amplification facilitator (product of gene 32 of bacteriophage T4) [30]. However, there is still considerable work to be done to improve detection by PCR from complex environmental matrices.

The results of the isolation of mycobacteria by culture provided new and exciting data. Seven mycobacterial species were identified from the positive cultures. By contrast, for two sites, herds 3 and 4, we obtained neither a positive culture nor a PCR signal. This reflects the difficulty of working with field samples for which there may be hazards that we do not control. Moreover, bacteriological cultures of these samples were impossible due to fairly frequent fungal contamination despite using an antifungal cocktail. More intensive sampling or better selection could improve the isolation of mycobacteria from environmental matrices and amoebae. The seven mycobacterial species identified (*Mycobacterium hiberniae*; *Mycobacterium nonchromogenicum*; *Mycobacterium icosiumassiliensis*; *Map*; *Mycobacterium arupensis*; *Mycobacterium parascrofulaceum*; *Mycobacterium genavense*) are already well-documented NTMs that affect farm animals and wildlife. NTMs, of which 61 species have been described in publications, are also opportunistic pathogens capable of causing lymphadenitis and infections of the lungs, skin, soft tissues, bursa, joints, tendon sheath and bones [31]. NTM poses two main problems: interference with the detection of bTB and other major mycobacteriosis such as JD; and the potential to cause significant or opportunistic infections leading to considerable economic losses [32]. The present field study clearly illustrates the impact of NTM in the environment (water point) of cattle could have on herd management in JD and bTB endemic zones.

The study also showed that *Map* is frequently detected in PCR and could be isolated in cultures, in accordance with previous work that has demonstrated that *Map* is widely distributed in the environment [7]. These results may also reflect the worrisome epidemiological situation of JD in France, with very high herd prevalence (around 50% for cattle farms) [1,2]. The isolation of *Map* by culture represents a significant challenge, especially with environmental samples containing many other microorganisms (up to 10^10^ genomes per gram of soil) [28] and many culture inhibitors. To support a hypothesis of disease transmission via environmental matrices, it is essential to demonstrate the presence of living bacteria. The detection of bacteria by PCR can be significantly overestimated (studies advance a factor of 100 as compared to culture) [33] since the detection of DNA can result from the presence of DNA traces or dead bacteria or even from dormant bacteria, which have been specifically observed for *Map* [34]. Our results of environmental *Map* detection in which bTB is also present seem to confirm the recent report of Byrne et al. [5], showing robust associations between bTB herd breakdown episode risk and concurrent *Map* infections at the herd level.

Although we could detect MTBC signature (IS6110, IS1081 and IS1561 positive qPCR) in samples, no strain of the MTBC could be isolated in culture despite animals infected with *M. bovis*. This result could suggest DNA persistence is superior to cultivable bacteria, which is in agreement with the observations of Adams et al. [29]. Even though France became free of bTB in 2001, zoonosis at the interface between animals and humans remains endemic in certain areas where wildlife is also affected. One of the reasons for this would be direct transmission by inhalation or ingestion of infected environmental substances [25,35,36]. In a previous study, *M. bovis* could be detected by PCR, but not isolated in culture, in environmental samples from tuberculosis outbreaks [28]. It seems that, similarly to our study, viable isolation of *M. bovis* from the environment is difficult to obtain. Therefore, sampling should target very recent, heavily affected outbreaks.

It was in a study by Delafont et al. [37] that, for the first time, an association between amoebae and mycobacteria, more precisely NTM, was observed in drinking water network, thereby highlighting the importance of FLA in the ecology of NTMs. More recently, screening of environments surrounding *Map*-infected herds led to the recovery of a free-living amoeba harboring a *Map* strain from a drinking trough [18]. In the present study, following the same approach, we were able to isolate and identify different amoebal isolates from all sampled herds. The discovery of this species diversity enlarges our knowledge of the environmental amoebae in interactions with livestock. All in all, 22 amoebal strains were isolated from the environment. They are distributed in four genera (*Naegleria*, *Acanthamoeba*, *Vermamoeba* and *Filamoeba*) and one group (Tubulinea). The majority of isolated amoebae belong to the *Naegleria* genus.

Interestingly, in herds 1 and 2, seven isolates of *Naegleria* were similar, and the best BLAST score was the same (KC164228.1), suggesting that this isolate is highly represented in the area, which is not the case for the other herds. Furthermore, *Acanthamoeba* and *Vermamoeba* are also highly represented. This was unsurprising as these are the major amoeba genera described in environmental water [9,38]. Finally, isolates from these genera have been characterised in interaction with *Mycobacterium*, particularly NTM, mainly identified in ponds and river water [7], suggesting that it could also occur in our environment [12].

## 5. Conclusions

Our study showed that a wide diversity of amoebae and NTM species, including *Map*, live closely in water troughs in herds monitored for bovine tuberculosis or JD. The exact association between amoebae and mycobacteria cannot be readily determined with these data. However, we hypothesize that given the results obtained elsewhere, *Map*, other NTM and *M. bovis* could survive in the environmental matrices of cattle thanks to the amoebae capable of hosting them. Therefore, the control of bTB and JD and sanitation of herds must consider environmental sources where *Map* and *M. bovis* could survive with amoebae and infect cattle and wildlife. In addition, numerous NTMs exposed to cattle may explain the cross-reactions observed during the diagnosis of these diseases. Therefore, the frequent presence of *Map* potentially associated with amoebae appears as an underestimated obstacle for controlling JD.

Further research is required to understand the association more clearly between mycobacteria and hosting amoeba from field samples, which could help to explain their survival in a multiplicative form and to more closely study the pathobionts consisting of a large variety of amoebae and mycobacteria.

## Figures and Tables

**Figure 1 animals-13-01781-f001:**
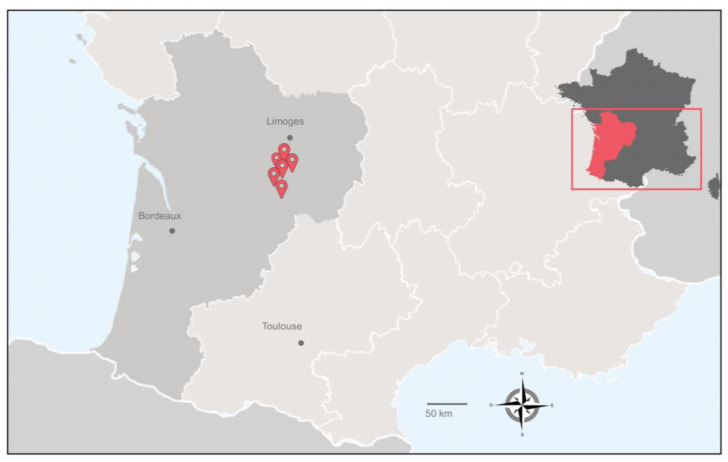
Location of the sampled bovine herds from southwest France Nouvelle Aquitaine region.

**Figure 2 animals-13-01781-f002:**
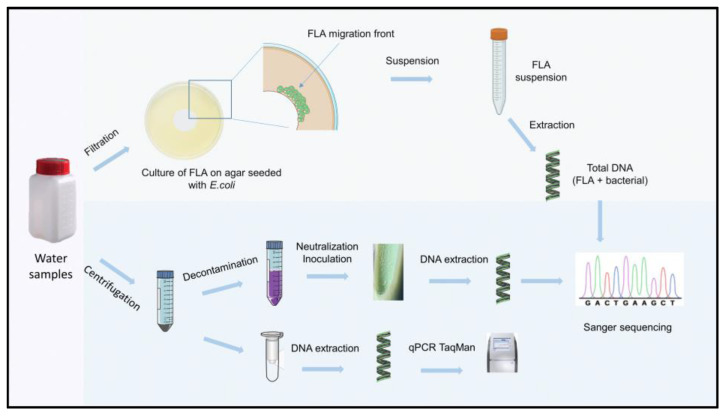
Experimental protocol for the analysis of water samples from cattle herds.

**Figure 3 animals-13-01781-f003:**
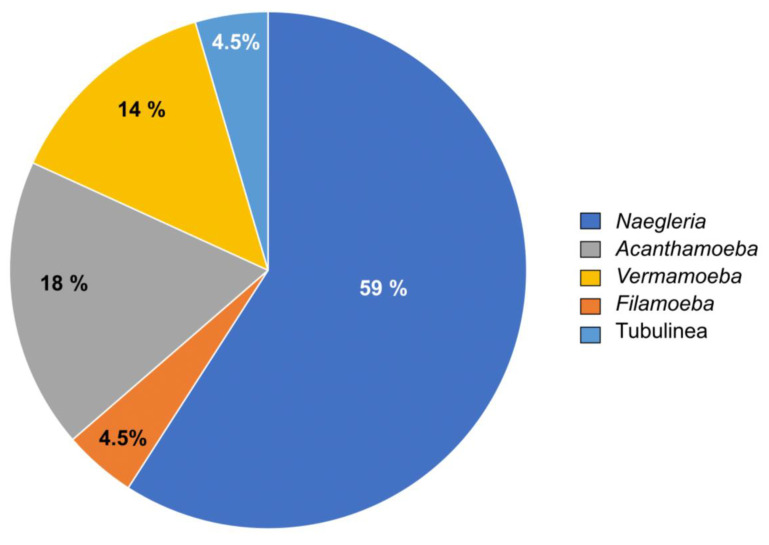
Pie chart of the frequency of isolated amoebic species.

**Figure 4 animals-13-01781-f004:**
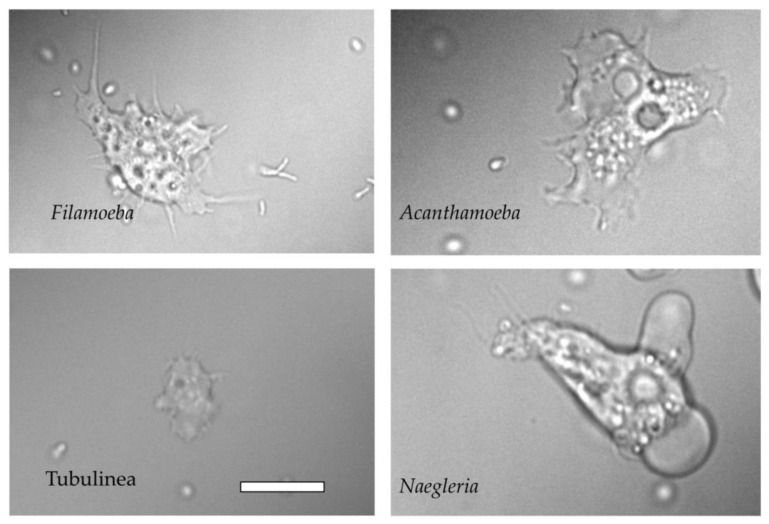
Morphological features highlighting the diversity of size and shape found in amoebae from the herd water sampling observed by differential interferential contrast microscopy. Bar length represents 10 μm for all panels.

**Table 1 animals-13-01781-t001:** Sampling details.

Herd N°	Number of Cattle	bTB Status	JD Status *	Sample Number	Nature of Water Sampling
1	68	positive 6 confirmed cows	ND	1	Pond
				2	Pond
				3	Pond
				4	Pond
2	82	positive Two confirmed cows	ND	5	Pond
				6	River
				7	Pond
				8	Pond
3	87	positive 1 confirmed cow	negative	9	Outdoor drinking trough
				10	Pond
4	19	positive Five confirmed cows	ND	11	Outdoor drinking trough
5	40	positive 12 confirmed cows	positive 3.5%	12	Outdoor drinking trough
				13	Outdoor drinking trough
				14	Outdoor drinking trough
				15	Outdoor drinking trough
				16	Outdoor drinking trough
				17	Outdoor drinking trough
6	235	positive 6 confirmed cows	positive 0.85%	18	Outdoor drinking trough
				19	Outdoor drinking trough
				20	Outdoor drinking trough

* ND; Not Determined, %; % of seroprevalence.

**Table 3 animals-13-01781-t003:** Mycobacterial species were identified by qPCR on water samples and by sequencing the *hsp* 65 gene after culture.

Herd N°	Sample N°	qPCR on Water Samples *	Culture ID
1 *	1	Ct > 40	*Mycobacterium hiberniae*
	2	Ct > 40	*Mycobacterium nonchromogenicum*
	3	Ct > 40	*Mycobacterium icosiumassiliensis*
	4	Ct > 40	*Mycobacterium nonchromogenicum*
2	5	Ct > 40	*Mycobacterium hiberniae*
	6	*Myco* sp.	*Mycobacterium avium* ssp. *paratuberculosis*
	7	Ct > 40	*Mycobacterium arupensis*
	8	Ct > 40	*Mycobacterium parascrofulaceum or genavense*
3	9	Ct > 40	Contaminated
	10	Ct > 40	Contaminated
4	11	Ct > 40	Contaminated
	12	*Myco* sp.; *Map*; MTBC	*Mycobacterium hiberniae*
	13	*Myco* sp.; *Map*; MTBC	Contaminated
	14	*Myco* sp.; *Map*; MTBC	Contaminated
5	15	Ct > 40	Contaminated
	16	*Myco* sp.; *Map*; MTBC	Contaminated
	17	*Myco* sp.; *Map*; MTBC	Contaminated
	18	Ct > 40	*Mycobacterium arupensis*
6	19	*Myco* sp.; *Map*; MTBC	Contaminated
	20	*Myco* sp.; *Map*; MTBC	Contaminated

* used primers described in Table 2 and targeted specific insertion sequences IS900, IS6110, IS1081 and IS1561 to identify Map or the MTBC species.

**Table 4 animals-13-01781-t004:** Characterization of environmental amoebae isolated from herd water samples.

Herd N°	Sample N°	Myco qPCR Signal *	T°	FLA Identification	Best Score ID	E Value
1	1	+	20	*Naegleria*	EU377592.1	0
			37	*Filamoeba*	GU320603.1	4 × 10^−154^
			37	*Naegleria*	KC164228.1	0
	3	+	20	*Acanthamoeba*	GU001160.1	0
			37	*Naegleria*	KC164228.1	0
2	5	+	20	*Naegleria*	KC164228.1	8 × 10^−156^
			37	*Naegleria*	KC164228.1	0
	6	+	20	*Naegleria*	KC164228.1	0
			37	*Naegleria*	KC164228.1	0
	7	+	37	*Vermamoeba*	KX856374.1	0
	8	+	20	Tubulinea group	FN562424.1	0
			37	*Naegleria*	KC164228.1	6 × 10^−132^
3	9	+	37	*Acanthamoeba*	AM408796.1	0
	10	+	20	*Naegleria*	OQ034614.1	0
			37	*Naegleria*	AY266314.1	0
4	11	+	20	*Acanthamoeba*	AY351644.1	0
			37	*Naegleria*	AY576367.1	0
5	12	+	37	*Vermamoeba*	KX856373.1	0
	14	+	37	*Naegleria*	MG699123.1	0
	18	+	37	*Naegleria*	KC164247.1	0
6	19	+	37	*Vermamoeba*	LC431240.1	0
	20	+	20	*Acanthamoeba*	HF930501.1	0

* qPCR atpE (*Mycobacterium* spp.) performed on water samples.

## Data Availability

Not applicable.

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
