# Peer review of "Presence of Non-Tuberculous Mycobacteria Including Mycobacterium avium subsp. paratuberculosis Associated with Environmental Amoebae"

_animals, 2023, doi:10.3390/ani13111781_

Round 1

Reviewer 1 Report

Was there an effort to determine whether viable mycobacteria were present in the amoeba? The presence of amoebae in drinking water isn't remarkable, nor would the presence of saprophytic mycobacteria. Likewise, finding MAP in drinking water on a farm where infected animals are present isn't surprising. 

Supplementary comments:

Without demonstrating that viable mycobacteria are present within the amoebae recovered from the livestock environment, there's nothing new here. Perhaps demonstrating that the recovered amoebae can engulf but not kill mycobacteria would bring us a bit closer to considering biological vectors as means of spreading MAP infection, and give the manuscript a bit more heft. 

Acceptable.

Reviewer 2 Report

The manuscript presents a very interesting topic of the presence of NTM and amoebae in the same environment. The methodology is very well chosen and written, but the manuscript needs correction in some aspects.

 The Results section must be improved as it is unclear.

 The Discussion mainly focuses on methodological aspects and should be extended to include epidemiological aspects.

 I also ask for standardization of naming and use of italics (e.g. in relation to Mycobacteria). After Figure 1 or Table 1 there should be a dot, not colon. I recommend a thorough review of the entire manuscript to standardize abbreviations, nomenclature, captions of tables and figures

 Further comments are presented below:

Line 1-4: I would consider changing the title. It may be worth changing the title to NTM in general.

Line 40-42: Keywords should be in alphabetical order with numbers next to them

Line 93: Was the aim really to test the role in disease transmission? It seems to me that this aspect has not been checked. Co-occurrence yes, but rather not transmission. 

Line 98: April 2018 to April 2019

Line 99: France (Figure 1 and Table 1)

Line 103: three

Line 116: Sample Number

Line 175 and 178: Please unify. There is past simple and present simple. 

Line 243: in Figure 1. The Table 

Line 250: Mycobacteria 

Line 264: Please add to the Discussion why do you think it was unusable to culture. MTBC is complex not species. I would change the naming in that Table generally, as it is not clear. 

Line 273-274: This is not a sentence corresponding to the results. 

Line 279: that these were water samples must be added in the title of the Table

Line 288: This sentence needs to be better worded. It is not known whether 4.5% was Vermamoeba or Vermamoeba and Tubulinea together. I added was 4.5% in brackets for each. What is more, Vermamooebea is two times in this sentence and there is no Filamoeba. I think the correct sentence would be “.. (14%), Filamobea (4.5%) and Tublinea (4.5%) (Fig. 3). ..”

Line 299: Together, it is not 100% and this Figure is not corresponding the manuscript. There is mistake for Filamonea and Tubulinea – there should be 4.5% for each them, not 1%. 

Line 325: water

Line 328-330: Please give appropriate reference. 

Line 346-348: It is the repetition of lines 338-339. 

Line 349: Map

Line 386: “no” is not necessary here as you did not use it before in the text. 

Line 395-409: Very nice conclusions!

Round 2

Reviewer 1 Report

Accept in present form